# Microcracks Reduction in Laser Hardened Layers of Ductile Iron

Eduardo Hurtado-Delgado *, Lizbeth Huerta-Larumbe 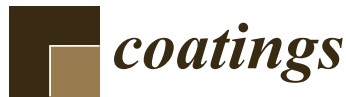, Argelia Miranda-Pérez 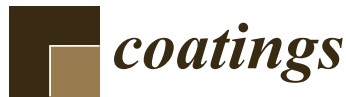 and Álvaro Aguirre-Sánchez 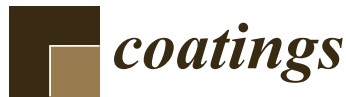

Corporación Mexicana de Investigación en Materiales, COMIMSA, Ciencia y Tecnología 790, Saltillo 25290, Coahuila, Mexico; lahuertal1990@gmail.com (L.H.-L.); argelia.miranda@comimsa.com (A.M.-P.); alvaroits@hotmail.com (Á.A.-S.)
* Correspondence: eduardoehd@alitmac.com; Tel.: +52-844-411-3200

**Abstract:** A study of surface hardening of Ductile Iron (DI) with and without austempering heat treatment was developed. The chemical composition of the material contains alloying elements such as Cu and Ni, that allow to obtain a Ductile Iron Grade 120-90-02, based on ASTM A536, which was heat treated to be transformed to Austempered Ductile Iron (ADI). Specimens of $10 \times 10 \times 5$ mm$^3$ were obtained for application of surface hardening by Nd:YAG UR laser of 150 W maximum power. The parameters used were advance speed of 0.2 and 0.3 mm/s and power at 105, 120, 135 and 144 W; the departure microstructures were fully pearlitic in the samples without heat treatment, and ausferrite for austempered samples. Microstructural characterization of hardened samples was performed were analyzed and martensite and undissolved carbides were identified in the pearlitic samples, while in ausferrite samples it was found finer martensite without carbides. The depth of hardened surface to different conditions and their respective microhardness were measured. The results indicate that the surface hardening via laser is a suitable method for improving wear resistance by means of hardness increment in critical areas without compromising the core ductility of DI components, but the surface ductility is enhanced when the DI is austempered before the laser hardening, by the reduction of surface microcracks.

**Keywords:** laser hardening; ausferrite; austempered ductile iron; nodular iron; heat treatment

## 1. Introduction

### 1.1. Characteristics of the Ductile Iron and the Austempered Ductile Iron

Ductile Iron (DI) is commonly used in many engineering applications, like sheet forming dies and rolling mills, as reported in literature [1,2]. Their high manufacturability and machinability represent an excellent combination of economic application performances [1,3,4]. By subjecting the DI to heat treatment, it transforms to Austempered Ductile Iron (ADI), which is essentially a spheroidal graphitic iron with ausferrite microstructure comprising mainly low carbon ferrite ($\alpha$) and high carbon retained austenite ($\gamma$). Because of an excellent combination of strength, ductility, toughness and fatigue resistance [5], ADI is now being increasingly used in key automobile components like crank shafts, steering parts, camshafts and gears [6–8], sometimes substituting steel parts [9].

### 1.2. Laser Surface Hardened Melting

Despite the good properties of DI and ADI, under some operating conditions such as erosive and corrosive environments its performance is limited by their relative low hardness [10–12]. This problem can be overcome by improving the surface properties of DI. High-power laser treatment (Nd: YAG, CO$_2$) is found as a significant technique to enhance the mechanical properties of ductile iron according to [13,14], including multipass and surface alloying using different powders [15,16], as reported by some authors who proposed some general guidelines for this process. Nevertheless, the presence of microcracks in the hardened case and other surface defects, as reported by [17], constrain

its applicability. Zheng et al. in 2013 proposed a novel technique in order to avoid crack formation in multiple overlapping laser tracks that represent a potential problem that must be reduced as possible [18].

To improve the wear resistance of the ductile iron, laser surface modification, without remelting, has been used in industrial applications, as it prevents failure by propagation of surface cracks [19,20]. In those cases when the iron has a ferrite-pearlite or even only a ferrite matrix, however, it is necessary to use laser hardening by melting the surface layer. This procedure creates a thin, microstructurally modified surface layer with a higher hardness. This layer consists of two parts: a melted zone and a heat hardened zone. The depth and width of the modified layer depend directly on the energy distribution and laser-beam diameter on the workpiece surface, the laser beam speed with respect to the workpiece, and the physical properties of the working material. The temperature and hence the properties of the surface can be controlled by the power density and the scan speed (in case of line hardening) or the interaction time (in case of single-shot hardening) of the beam [21].

Laser surface hardened melting (LSM) offers several advantages over other surface-modification techniques. This process is a non-equilibrium method of surface modification reaching cooling rates around 103–108 K/s, which are considerably high. The resultant microstructures, some of them metastable phases, are mainly composed of unique properties that are only obtained with this process and not with conventional ones [22,23]. Laser hardening is usually constrained to low heat inputs in order to avoid surface microcracks in ductile irons, resulting in shallow hardened layers [24].

### 1.3. Microstructure and Typical Hardness

During laser hardening process, the surface of the irradiated material is heated in order to transform the microstructure of the heat affected zone into austenite. The surrounding material acts as an efficient heat sink, quickly cooling the material likely below the martensite start temperature [19].

The present microstructure after treatment LSM depends on the parameters used in the process as well as the initial microstructure of the workpiece. In this way, Benyonius [10] reported that if the microstructure in the DI is ferritic, after LSM treatment, eventually it will be formed a microstructure of dendrites made of austenite, surrounded by continuous networks of $Fe_3C$ and some martensite needles within the austenite islands. Alabeedi [23] presented an LSM treatment in a ferritic DI and showed that the laser melting led to complete dissolution of the graphite nodules which on solidifying created an interdendritic network of ledeburite eutectic with a very fine structure, good homogeneity and high hardness (650 HV). In another paper presented by Fernández [2], it was studied the effects of laser surface treatment on the microstructure, crackability and stresses generated on laser hardened layers produced in several ductile cast iron materials; in the study, two principal types of spheroidal graphite were selected. Considerable cracking by thermal stresses was produced on both irons, pearlitic and acicular bainitic, the energy densities achieved was above 40 J/mm$^2$. It was observed that lower energy densities refrain cracking but only in the pearlitic ones, this was achieved by the excessive austenite retention that controlled the generation of transformational stresses. Grum [25] reported that in case of having an initial pearlitic matrix in the ductile iron, after a laser surface hardening, the resultant and predominant structure produced in the surface is martensite.

Regarding ADI that presents ausferrite microstructure initially, Roy [26] observed that the structure of the laser surface melted area was mainly austenitic, while a higher microhardness of more than 1000 HV happened with a martensitic microstructure. Furthermore, LSM produced more compressive residual stresses and enhanced significantly the wear resistance of the austempered ductile iron [19]. Putatunda [27], who applied laser hardening techniques, carried out an investigation on ADI. They used manganese phosphate coatings and colloidal graphite to achieve more uniform hardness. The hardness

values reached were around 700 HV and the microstructure of a thin hardened martensitic layer improved the mechanical properties of the material.

Amirsadeghi [28] studied the microhardness and wear resistance of different microstructures formed by tungsten inert gas (TIG) surface melting and chromium surface alloying (using ferrochromium) of ADI. Surface melting resulted in the formation of a ledeburitic structure in the melted zone, and this structure has hardness up to 896 HV, as compared to 360 HV in that of ADI. The results also indicated that surface melting reduced the wear rate of the ADI by approximately 37%. Finally, in a work presented by Grum [25], it was studied the laser surface-melt hardening in gray and nodular irons, and found that the melting produced by low-power laser beam can obtain an adequately modified hardened layer, which results in an increment of the surface wear resistance. Material properties play a dominant role in determining the interaction between the laser beams and engineering materials. Many material properties change with temperature. The mechanical properties of many engineering materials may be favorably modified by application of a suitable heat treatment, which can be full or superficial [29]. One of the most important superficial treatments of metals has been the laser transformation hardening of steel [29,30], but this treatment can also be applied successfully to ductile irons.

In the present study, an attempt has been made to enhance surface hardness and wear resistance of DI with and without austempering heat treatment. The aim of this work is to show that the austempered heat treatment before laser hardening of ductile iron is effective in reducing the amount of surface microcracks in a wide range of heat inputs.

## 2. Materials and Methods

### 2.1. Ductile Iron

The nodular iron utilized for these experiments corresponds to 120-90-02 grade, under the ASTM A536 [31] standard and it has a chemical composition that is typical for this type of irons. Cu and Ni were added to increase the amount of pearlite in the as-cast microstructure. The chemical composition is shown in Table 1.

**Table 1.** Chemical composition (wt. %) of the Ductile Iron (DI).

| C | Si | Mn | Cu | Mo | Ni | Mg | Cr | P | Fe |
|------|------|------|------|------|------|------|------|-------|------|
| 3.52 | 2.11 | 0.32 | 0.39 | 0.21 | 0.33 | 0.05 | 0.15 | 0.025 | Bal. |

The carbon equivalent for this ductile iron was of $C_e = 3.82$, which is defined as hypereutectic iron. Besides this, eutectic saturation was calculated as $S_c = 1.05$ considering Si, Mn and P, according to the equation presented in [32].

### 2.2. Austempered Ductile Iron

The DI samples were fully austenitized at 900 °C for 120 min and austempered in an isothermal salt bath at 340 °C to 360 °C for 60 min followed by cooling in air at room temperature. The salt bath was 60% $KNO_3$ and 40% $NaNO_2$ a schematic diagram of the austempering heat treatment process is shown in Figure 1.

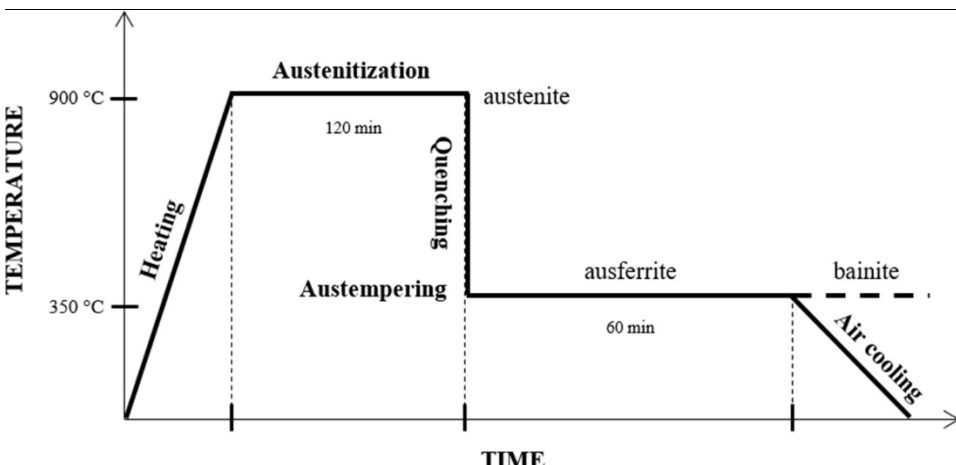

**Figure 1.** Schematic heat treatment process.

### 2.3. Experiment Design and Laser Parameters

In this study, DI samples with and without austempering heat treatment were surface hardened by UR LaserTechnologie Nd:YAG laser of 150 W maximum power. Diameter of laser spot was about a half of the bead width, which can be observed as the length of the transversal fusion zone near the surface. It is difficult to assign a defined value because there is a power density reduction from the center to de periphery, following a Gaussian function. However, the effective spot size, in this case can be considered as that which produces the fusion the metal in the axial direction; it is approximately the full length of the fusion zone at the half depth, shown in the micrographs in results section. The specimen dimensions were 10 × 10 × 5 mm. The selected parameters for the surface treatment were the following:

- Two advance speeds: 0.2 and 0.3 mm/s
- Four powers: P1 = 144 W, P2 = 135 W, P3 = 120 W, P4 = 105 W

Four samples were used for the experiment: two DI and two ADI samples. For each sample, four laser-melted beads were produced, one for each power level, according to the Figure 2 schemes.

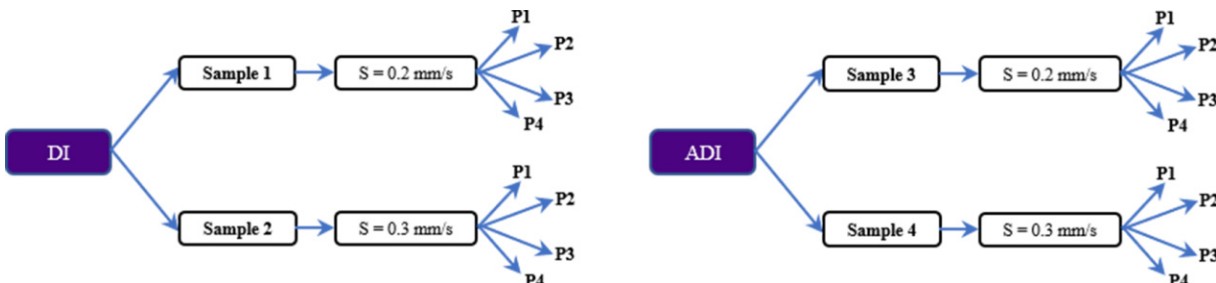

**Figure 2.** Design of experiments for the laser surface hardening.

### 2.4. Laser Surface Melting-Hardening of DI and ADI

Cross sections of the samples were cut for metallographic examination. The microstructural characterization was consisted in grinding (using 120, 240, 320, 600, 800 SiC paper) and polishing with a one μm diamond paste, using Nital at 1% as reactive etchant for 5 s. The samples were inspected in the different seams weld zones with an optical microscope (OM) Nikon Eclipse MA200 and electron microscope Tescan Mira 3; besides this, hardness examinations in each zone were performed using 300 g for Vickers indentation. Microhardness evaluation, using a Wilson hardness Tukon 2500 equipment, was performed in order to compare the parameters effect in the weld beads and to find a proper

combination with the higher hardness without cracks. Wear resistance is favored with these characteristics [33]. Figure 3 shows each zone area where RZ and HZ correspond to the re-melted zone and hardened zone (affected by the heat), respectively. All the indentations for microhardness profile are shown as well: 3 indentations for each position (H1, H2, H3, H4) in order to obtain the average and more reliable results.

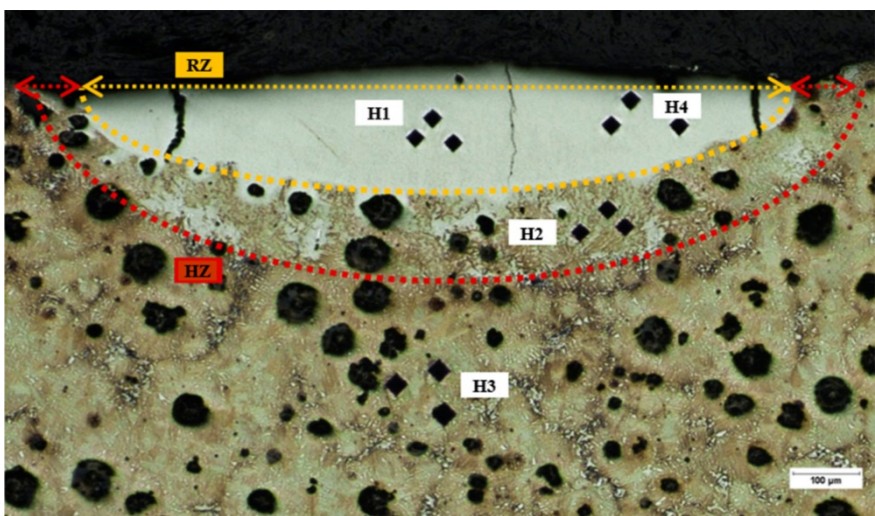

**Figure 3.** Current zones in the weld seam of the laser surface hardened melting (LSM).

## 3. Results and Discussion

### 3.1. Base Material

The original microstructures for DI samples before the surface hardening are shown in Figure 4. Microstructure consist mainly of pearlite and graphite nodules, around 30 µm diameter, with a composition of 84% and 12%, respectively, and the difference may indicate segregation zones predominantly Ni and Cr (Figure 4a). The microstructure was measured by image analysis with Image Pro Software and NIS Element coupled to the OM; for the ADI samples (Figure 4b), it was present ausferrite, graphite nodules and austenite islands, and the segregated zones disappeared with the austempering. The compositions of these phases were 83%, 11% and 6%, respectively, measured by image analysis.

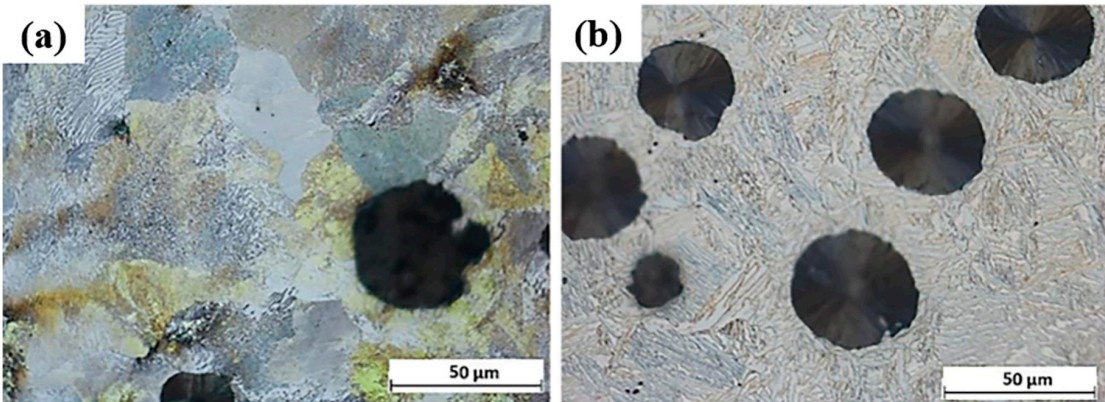

**Figure 4.** Optical micrograph, (**a**) DI (Perlite + Nodules + Segregations), (**b**) Austempered Ductile Iron (ADI; Ausferrite + Nodules + Austenite).

On average hardness in the DI was 295 HB and heat treatment increased the ADI to 314 HB.

## 3.2. Dimensions and Morphology of the Melted Zone

As the iron castings are a mixture of phases of iron and graphite, when the metal is re-melted by the laser it dissolves all the free graphite in the liquid and in the subsequent and fast cooling it results in an oversaturated carbon alloy, mainly formed by martensite, some retained austenite and iron carbides. The proportion of these phases depends upon the maximum temperature reached, the holding time at this temperature and the cooling rate. The first variable depends on the laser power, and the last two variables depend on the advanced speed of the laser beam and the thermal metal properties.

The DI can be hardened by a laser beam because of the great amount of carbon contained in the microstructure which can be dissolved and form martensite after the fast fusion of the metal, taking advantage of the high-density power of the laser beam [34]. This can be realized using even a low power equipment, of only 150 W, like that used in this study. In Figure 5 shows the morphology of the left half beads produced with the laser at its highest power, 144 W and their dimensions. DI beads are wider and deeper than their corresponding ADI seams with slower speed. They also have more and larger surface cracks. DI cracks are observed as larger as the hardened layer, as in S1P1 where its value corresponds to 371 μm, crossing thoroughly the melted zone. Cracks in the hardened iron are in the diagonal and vertical direction. As reported for surface alloyed carbon steels [35,36], the presence of cracks is due to hot cracking. Since susceptibility for hot cracking is determined by the alloy plasticity and solidification temperature range (ΔT); in the surface area, a composition near the eutectic point (3.4–4.5 C wt.%) is expected, so ΔT is small, resulting in some plasticity and fine dendritic structure at laser temperatures. Therefore, once the metal is partially solid, crack appearance depends on the thermal contraction of the remaining molten metal.

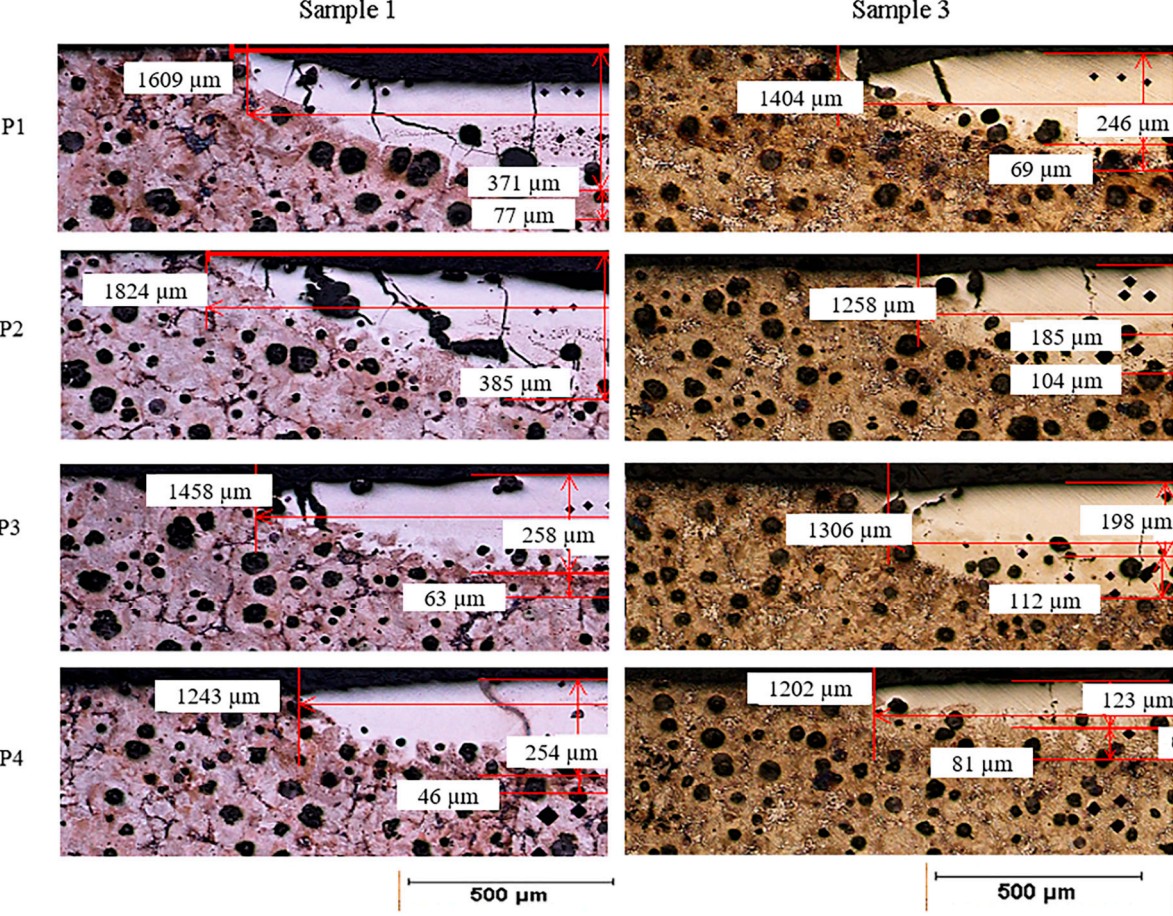

**Figure 5.** Seams dimensions of DI: Sample 1 (S1) and ADI: Sample 3 (S3), with 0.2 mm/s speed.

If the advance speed of the laser source is changed from 0.2 mm/s to 0.3 mm/s, DI beads are wider than their corresponding ADI beads, in all cases, but not necessarily deeper; this can be observed in Figure 6, where the depth of ADI increases except for the highest power. Moreover, due to their greater depths of melted zone, 0.2 mm/s speed presented greater nodule dissolution.

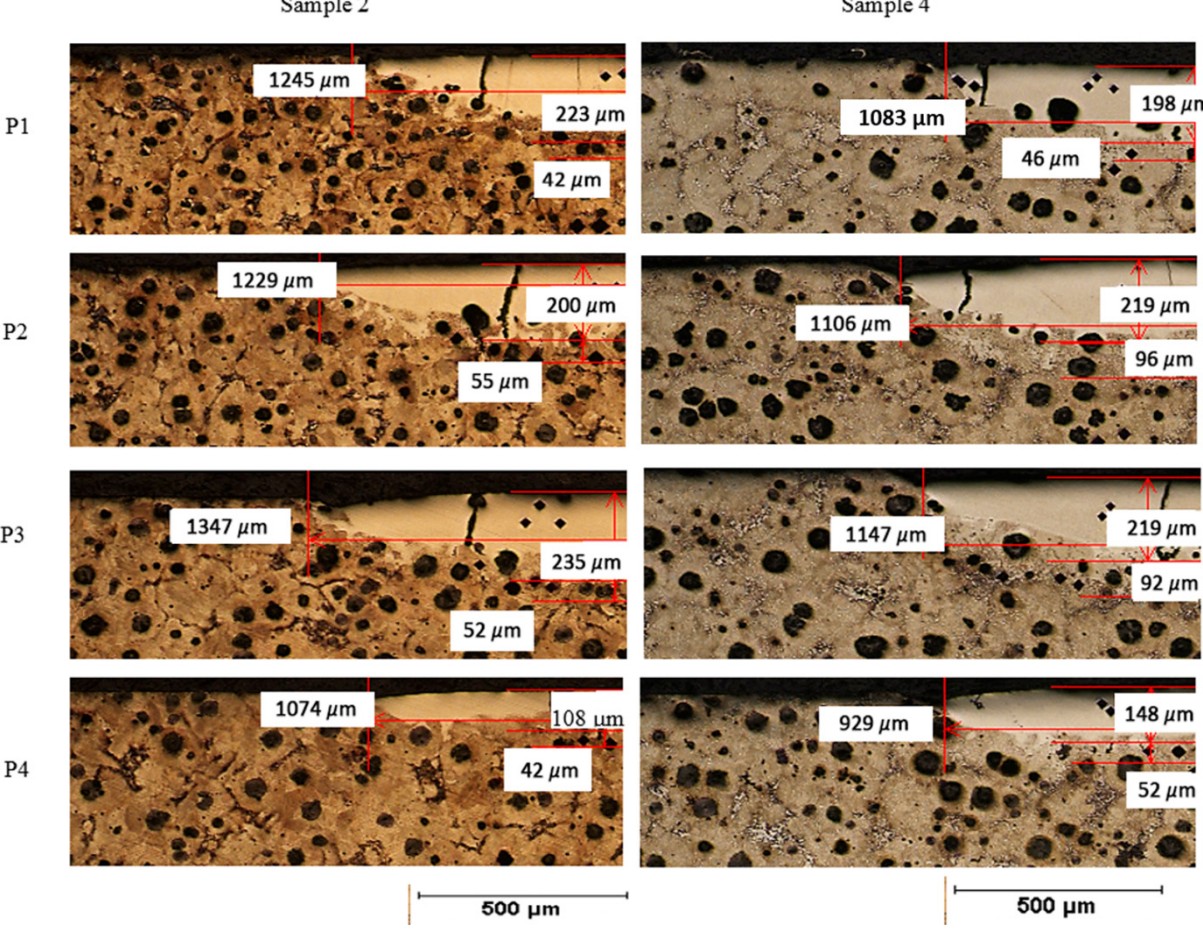

**Figure 6.** Beads dimensions of DI: Sample 2 (S2) and ADI: Sample 4 (S4), with 0.3 mm/s speed.

The results shown in Tables 2 and 3 indicate that the power input density was between 325 J/mm$^2$ and 579 J/mm$^2$ for speed of 0.2 mm/s; furthermore, the power density was between 248 J/mm$^2$ and 385 J/mm$^2$ for 0.3 mm/s. This outcome is far from the power density of 40 J/mm$^2$, reported by [2]. Nevertheless, there is no evidence of crack appearance at the lowest power used in the experiments of this work, so the later reference value is likely conservative.

**Table 2.** Bead measures in different samples (0.2 mm/s speed).

|  | S1_P1 | S1_P2 | S1_P3 | S1_P4 | S3_P1 | S3_P2 | S3_P3 | S3_P4 |
|---|---|---|---|---|---|---|---|---|
| **Width (µm)** | 1609 | 1824 | 1458 | 1243 | 1404 | 1258 | 1306 | 1202 |
| **Depth melting (µm)** | 371 | 385 | 258 | 254 | 246 | 185 | 198 | 123 |
| **Thickness HAZ (µm)** | 77 | >10 | 63 | 46 | 69 | 104 | 112 | 81 |

**Table 3.** Bead measures in different zones (0.3 mm/s speed).

|  | S2_P1 | S2_P2 | S2_P3 | S2_P4 | S4_P1 | S4_P2 | S4_P3 | S4_P4 |
|---|---|---|---|---|---|---|---|---|
| **Width (µm)** | 1245 | 1229 | 1347 | 1074 | 1083 | 1106 | 1147 | 929 |
| **Depth melting (µm)** | 223 | 200 | 235 | 108 | 198 | 219 | 219 | 148 |
| **Thickness HAZ (µm)** | 42 | 55 | 52 | 42 | 46 | 96 | 92 | 52 |

### 3.3. Microstructure

### 3.3.1. Base Metal

Figure 7a shows the pearlitic initial microstructure of the DI samples; the pearlite is constituted of $Fe_3C$ lamellae with ferrite, and the hardness value is around 320 HV. In Figure 7b, the ausferritic microstructure that is characteristic of ADI is presented; this structure has a hardness of approximately 410 HV and consists of high carbon austenite plates with ferrite. Ausferrite microstructure refinement is dependent of the austempering temperature. In this case, the ausferrite is not as fine as could be under lower austempering temperatures [37].

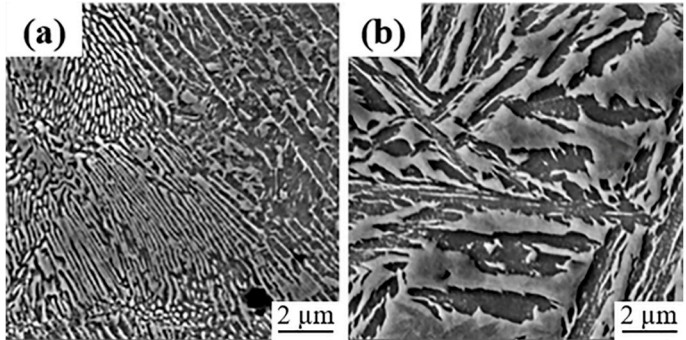

**Figure 7.** Microstructure, (**a**) DI: Pearlite, (**b**) ADI: Ausferrite.

### 3.3.2. High Power Remelted Zone

On the fusion zone (H1) from DI samples, sample 1 (S1) presented in Figure 8a has a large amount of refined acicular $Fe_3C$ (AFC), high carbon martensite (HCM) at the bottom and some retained austenite ($\gamma_r$); for that reason, the reached hardness in this area was >1000 HV. On the other hand, sample 2 (S2) obtained a lower hardness (731 HV). As can be seen, the hardness reduction from S1 to S2 is due to the increase of speed, $Fe_3C$ is present in lower quantities and is coarser (ACC), the presence of HCM increased as well as $\gamma_r$, and the carbides decreased and some of them were separated from martensite to be grouped and form a platelike constituent (Figure 8b).

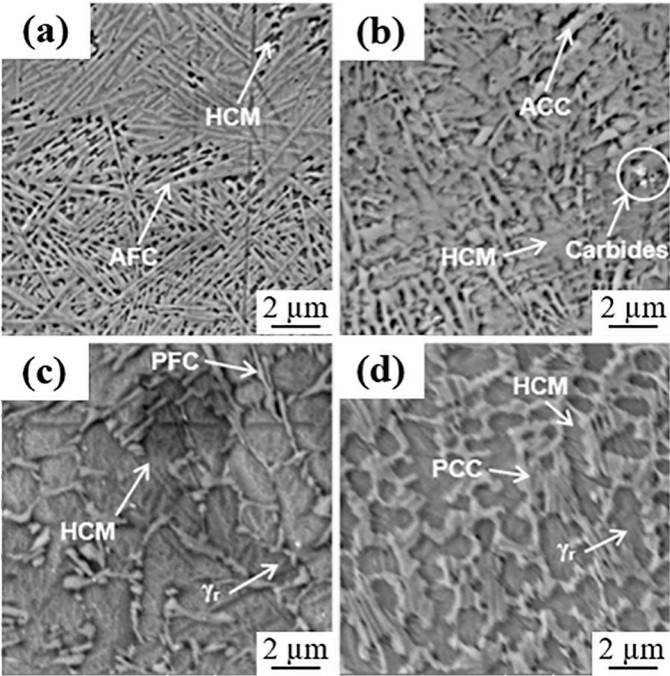

**Figure 8.** Microstructure of the melting zone (H1) to 150 W (P1), (**a**) Sample 1, (**b**) Sample 2, (**c**) Sample 3 and (**d**) Sample 4.

Regarding the ADI, the reached temperatures on the fusion zone were lower than those for the DI; for that reason, the dissolved carbon amount was minor. Figure 8c shows the H1 zone microstructure which corresponds to sample 3 (S3); it is constituted of HCM islands surrounded by $Fe_3C$ fine plates (PFC) and small amounts of $\gamma_r$, with almost 968 HV. This microstructure was formed due to the lower laser speed and the higher temperature reached; the $\gamma$ from the sample was not rich in carbon, which prevented its stabilization at room temperature and, from the rapid cooling, the microstructure transformed mostly to HCM [34,38] and the surrounded liquid to $Fe_3C$.

At higher advance speed, the maximum temperature in the sample is lower and the arising $\gamma$ dissolves more carbon that stabilizes it at room temperature. Figure 8d exhibits sample 4 (S4), on the H1 zone, and presents large quantities of $\gamma_r$ in form of islands surrounded by coarser and greater $Fe_3C$ plates (PCC). Some zones, wherein $\gamma$ could not dissolve too much carbon, transformed into HCM with rapid cooling. The higher amount of $\gamma_r$ in this sample is the cause of the hardness decay (864 HV) compared to S3.

Using the diagram in Figure 9, the microstructures from different samples were deducted. In this diagram [39] the carbon concentration curves as a function of the cooling and heating rates are presented, as an example. The amount and type of resultant microstructures depend on the maximum temperature reached, as well as the heating and cooling rates, which in turn depend on the power and advance speed of the laser beam. This can be visualized using the Fe-C-Si phase diagram, the heating and cooling cycles superimposed for a high speed-high power, and a low speed-low power beams, both acting upon a surface of nodular iron with a fully pearlitic matrix (i.e., 0.8% C). In the first case, fusion zone, a high temperature is reached very fast, but there is no time to dissolve a great amount of carbon from the graphite spheroids; after the fast heating, a fast cooling is followed and the result is an austenite with low carbon content which is transformed to a mixture of martensite and some $\gamma_r$, surrounded by $Fe_3C$, which arose from the molten metal, as the final microstructure. In the second case (low speed-low power), the reached temperature is lower, but there is more time to dissolve carbon and, according to the Fe-C-Si diagram, carbon has more solubility in austenite at lower temperatures above the eutectic; after heating, the cooling is faster but, because the greater content of carbon, less austenite

transforms to martensite, since $M_s$ (martensite start transformation temperature) is lower, and the austenite is more stable at ambient temperature in this condition. The total amount of molten metal can be greater, at lower speed (and lower T), but the liquid volume fraction is lower because, at lower temperature, more austenite can coexist with the liquid; the carbon dissolution has the effect to lower the liquidus temperature in the Fe-C-Si system. Consequently, the volume fraction of cementite formed at the end is greater when higher advance speed is used, even when the temperature had been lower. It is important to know this, because the number and extension of surface microcracks depend on the amount of martensite and cementite in the microstructure.

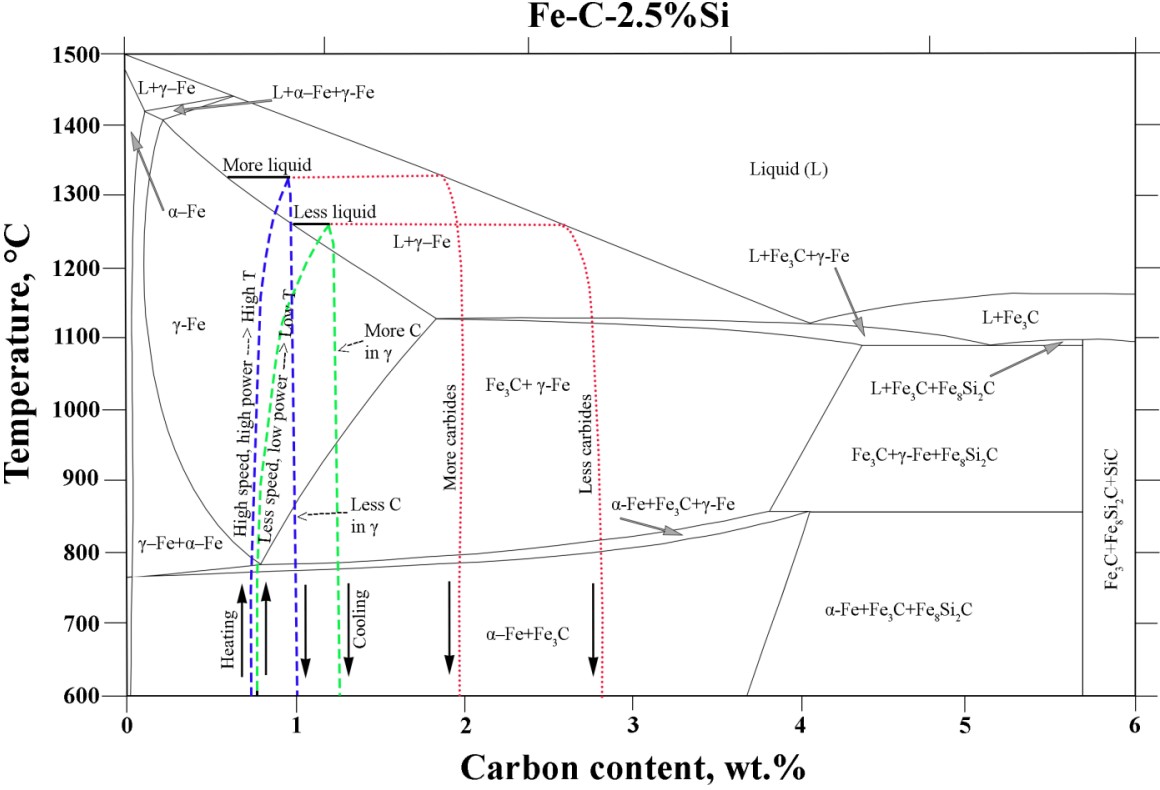

**Figure 9.** Schematic Fe-C-Si diagram [39].

### 3.3.3. Low Power Remelted Zone

Due to the low temperature reached on the H1 zone (P4) in S1 sample the microstructure formed (Figure 10a) consisted mostly in $\gamma_r$ with HCM islands surrounded by PFC; the amount of carbides decreased compared to the P1 power, and for that reason the hardness decay >1000 HV (P1) to 633 HV (P4). In Figure 10b are shown the microstructures obtained in the H1 zone from S2 sample; unlike the previous one, more presence of PFC is evident since the required liquidus temperature in this zone was greater than S1 at the same power, due to the lower carbon dissolution.

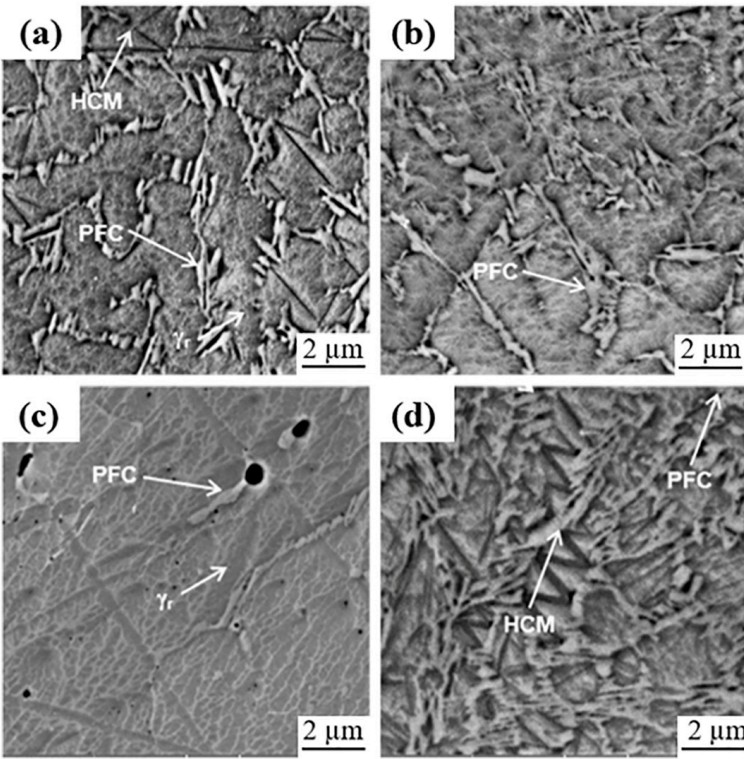

**Figure 10.** Melting zone (H1) microstructure 70 W (P4), (**a**) Sample 1, (**b**) Sample 2, (**c**) Sample 3, (**d**) Sample 4.

Instead, for ADI at P4 power with 0.2 mm/s the maximum temperature barely melted the metal; it exceeded the liquidus line but did not get to dissolve a considerable amount of nodules; when cooling begins, the material transforms to austenite with high carbon content and a small quantity of liquid, at certain intermediate temperature this small quantity of liquid transforms to ledeburite and high carbon austenite. A slight amount of $Fe_3C$ that encloses the $\gamma_r$ grains arises from ledeburite. The sample at higher speed S4_P4 did not reach the liquidus line when it started to cool; the liquid that forms, as well as the austenite, has lower carbon content than the maximum achievable, which corresponds to the eutectic, just at the maximum solubility of austenite; at high cooling rate the liquid rapidly reaches an inferior temperature than the eutectic, starting the cementite formation. Most of the prior austenite, with insufficient carbon to prevail as a metastable phase at low temperature, transforms to martensite.

### 3.3.4. HAZ

As observed in Figure 11, the heat affected zone is very similar for both materials and both speeds; it consists mainly of martensite and retained austenite as mentioned in [16]. The structure is finer for the DI samples because it reached lower temperature and the carbon homogenization is faster in the pearlitic condition than in the ausferritic.

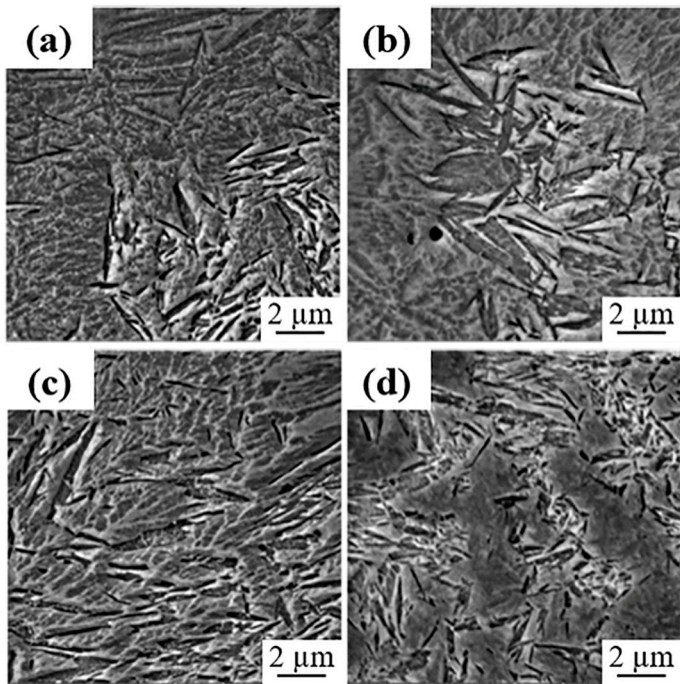

**Figure 11.** Microstructure of the HAZ (H2) to 70 W (P4), (**a**) Sample 1, (**b**) Sample 2, (**c**) Sample 3, (**d**) Sample 4.

### 3.4. Cracks and Microhardness

### 3.4.1. High Hardness

Table 4 shows microhardness results obtained at different positions of all samples. The beads of DI that presented higher hardness values were those with the lower speed and higher power (P1), this is due to most of the microstructure being $Fe_3C$ in S1_P1; the $Fe_3C$ was in acicular form and martensite in small amounts. For that reason, high hardness was obtained >1000 HV, so that it caused embrittlement at cooling producing a great number of cracks (Figure 5). On the other hand, S2_P1 presented a coarser $Fe_3C$ with larger martensite volume fraction than S1_P1; therefore, the hardness resulted lower (731 HV) and the cracks presence was reduced, hence the sample was less fragile, as pointed out in [2].

**Table 4.** Microhardness at different positions (HV).

|  | **H1 (HV)** | **H2 (HV)** | **H3 (HV)** | **H4 (HV)** |
|---|---|---|---|---|
| **S1_P1** | 1022 | 846 | 335 | 1055 |
| **S1_P2** | 1145 | 748 | 268 | 1017 |
| **S1_P3** | 872 | 529 | 327 | 673 |
| **S1_P4** | 633 | 606 | 363 | 565 |
| **S2_P1** | 731 | 397 | 317 | 750 |
| **S2_P2** | 695 | 410 | 318 | 563 |
| **S2_P3** | 738 | 610 | 314 | 780 |
| **S2_P4** | 662 | 571 | 305 | 765 |
| **S3_P1** | 968 | 560 | 415 | 908 |
| **S3_P2** | 557 | 573 | 424 | 623 |
| **S3_P3** | 609 | 561 | 358 | 648 |
| **S3_P4** | 542 | 674 | 401 | 565 |
| **S4_P1** | 867 | 569 | 308 | 713 |
| **S4_P2** | 724 | 647 | 413 | 643 |
| **S4_P3** | 898 | 606 | 353 | 939 |
| **S4_P4** | 721 | 576 | 394 | 667 |

At the same P1 in the ADI samples, S3 achieved hardness values of 968 HV that are close to S1_P1, the crack appearance was abruptly reduced due to the microstructure but graphite flotation was evident on the surface. In S4_P1, the hardness value was 867 HV since $\gamma_r$ transformation, compared to S3_P1, the cracks were smaller and graphite flotation was removed.

### 3.4.2. Low Hardness

At lower power and speeds, the hardness decreased significantly from 1022 HV (S1_P1) to 633 HV (S1_P4), and crack presence was eliminated due to the resulted microstructure. At higher speed hardness, decay was not evident for beads P1 to P4, since it passes from 731 HV to 622 HV. In the ADI at lower power a similar behavior was maintained as in DI samples, since hardness decrease was more noticeable at lower speeds (968 HV a 542 HV); this behavior was also reported in [19].

### 4. Conclusions

The parameters which demonstrate improved performance regarding hardness, dimensions and crack formations were:

- Material = ADI
- Power = P4 (105 W)
- Speed = 0.3 mm/s

Surface hardening by laser treatment is reliable, but cracks are generated during the solidification if not properly applied.

ADI re-melted beads are more narrow than their corresponding DI beads.

The highest hardness was 1145 HV obtained from the DI condition, without austempering heat treatment.

The DI samples presented more and larger cracks in all experimental conditions because the contraction of cementite during cooling.

ADI is less prone to crack formation than DI, because it contains less cementite and more martensite, and because ausferrite is more heat-conductive than pearlite.

Concisely, this work has demonstrated with no doubt that the ADI is a better option for laser hardening than DI, because the former can dissipate the heat input faster and more evenly, due to the thermal characteristics of both materials, identical in chemical composition, but not in phase composition. ADI has a metal conductive matrix with carbon saturated austenite and graphite nodules, while DI has a mixture of metal and ceramic (pearlite) matrix and graphite nodules which is less heat-conductive. Microcracks are related to the excessive accumulation of heat, which produces higher thermal gradients and formation of greater amounts of carbides.

**Author Contributions:** E.H.-D. conceived and planned all experiments. L.H.-L. and Á.A.-S. performed all experimentations. A.M.-P. and L.H.-L. wrote the manuscript, and all authors participated in results analysis and discussion. All authors have read and agreed to the published version of the manuscript.

**Funding:** The present work was funded by CONACyT Mexico under the project Proinnova 216536.

**Institutional Review Board Statement:** Not applicable.

**Informed Consent Statement:** Not applicable.

**Data Availability Statement:** Data discussed in this contribution is available on request from the corresponding author.

**Conflicts of Interest:** The authors declare no conflict of interest.

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
