# Peer review of "Microcracks Reduction in Laser Hardened Layers of Ductile Iron"

_coatings, doi:10.3390/coatings11030368_

Round 1

Reviewer 1 Report

I have placed my comments in the attached file.

Reviewer 2 Report

Dear Authors,
I have read your paper "Microcracks reduction in laser hardened layers of ductile iron" carefully.
This paper describes the effect of the austempering heat treatment and laser hardened on the microcracks in the surface layer. 
The paper is easy to read. But in the introduction, there is artifacts line 122-130
The methods are not properly described, so that other research groups may not reproduce them.
The paper is interesting. However, it requires few corrections.
1. Line 149. Please correct.
2. Please, add more information about laser hardening treatment, for example, the diameter of the spot, etc.
Please, add more information about test equipment. It’s recommended to add the paragraph with the type of the optical and electron microscopes, test machines for wear test, and microhardness, and Brinnele hardness of this work to the second section.
3. I did not find the information about the size of the cracks.
4. I did not find the information about the wear-resistant test. Please, add more results.
5. Please specifically discuss the advantages of your work. 
6. The conclusion of this work contains the list of the obtained results. What's new in surface hardening has been identified? In the Conclusion part, there is a lack of highlighting a novelty of the present work.  

Authors should carefully study the comments and make improvements to the article step by step. After major changes can an article be considered for publication in the "Coatings"

Reviewer 3 Report

1Many weak formulation and grammar mistakes please correct them

2Please correct this formulation “samples was performed where coarse martensite and undissolved carbides were identified”

3From here “The introduction should…to ….further details on references” should be removed

4Please highlight better the novelty in terms of scientific understanding

5Please cite this standard “ASTM A536 standard” as they differ from year to year

6Weak formulation “and it has a chemical composition that is typical for this type of irons”

7Please use only English “0.2 y 0.3 mm/s”

8The selected parameters have any physical meaning ?

9I suspect you have used for batch of samples not four in total cause like that there is no any statistical meaning

10Please describe in details the sample preparation procedure “grinding and polishing” is no possible to replicate if I use like that

11Which software did you have used for “image analysis”?

12Not clear “In this case, is not as fine as could be under lower temperature”

13From line 248 you describe the FeC diagram but no indication to actual your material, this is well know and only make sense if you discuss it against your material

14“The highest hardness is obtained with DI as-cast” Ok but it is much better if you provide some value

15“This section is not mandatory but can be added to the manuscript if the discussion is unusually long

or complex.” I suggest removing this sentence cause make no sense

16Most of the references are old which can lead to no novelty, therefore I suggest inserting some more new references

Round 2

Reviewer 2 Report

Dear Authors,

I have read your modified paper "Microcracks reduction in laser hardened layers of ductile iron" carefully.

The materials and methods are properly described, so that other research groups may reproduce them. Explanations are clear and the paper is easy to read.

I can recommend the Editor to accept this revised manuscript to be published in Coatings.

Reviewer 3 Report

Thank you